# LESS IS MORE: SELECTING INFORMATIVE AND DIVERSE SUBSETS WITH BALANCING CONSTRAINTS

## ABSTRACT

Deep learning has yielded extraordinary results in vision and natural language processing, but this achievement comes at a cost. Most models require enormous resources during training, both in terms of computation and in human labeling effort. We show that we can identify informative and diverse subsets of data that lead to deep learning models with similar performance as the ones trained with the original dataset. Prior methods have exploited diversity and uncertainty in submodular objective functions for choosing subsets. In addition to these measures, we show that balancing constraints on predicted class labels and decision boundaries are beneficial. We propose a novel formulation of these constraints using matroids, an algebraic structure that generalizes linear independence in vector spaces, and present an efficient greedy algorithm with constant approximation guarantees. We outperform competing baselines on standard classification datasets such as CIFAR-10, CIFAR-100, ImageNet, as well as long-tailed datasets such as CIFAR-100-LT.

## 1 INTRODUCTION

Deep learning has shown unprecedented success in many domains, such as speech (Hinton et al., 2012), computer vision (Krizhevsky et al., 2012; Szegedy et al., 2015; He et al., 2016), and natural language processing (Sutskever et al., 2014; Devlin et al., 2018). This success generally relies on access to significant computational resources and large human-annotated datasets. For example, energy consumption and carbon footprint of developing common NLP models are shown to be comparable to the lifetime emissions of a car (Strubell et al., 2019). Similarly, human annotation is also a time-consuming and expensive process; Badrinarayanan et al. (2010) reports that semantic labeling of a single video frame takes around 45-60 minutes.

In this paper we ask the following question: Given a large unlabeled dataset along with a small labeled (seed) dataset and an annotation budget, how do we select a subset of the unlabeled dataset, which, when annotated, will achieve the best performance? While this is addressed by many classical active learning methods (Settles, 2009), in this work we focus on the modern deep learning setting with CNNs and very deep networks such as ResNets (He et al., 2016). Typical active learning methods employ an iterative process where a single image is labeled and used to update the model at each step. The updated model is then used to pick the next image and so on. Such an approach is not feasible for deep networks. We therefore choose to study this problem in a batch setting (Sener & Savarese, 2018; Zhdanov, 2019; Shui et al., 2020; Kim et al., 2021; Ghorbani et al., 2021).

In this setting, we first train an initial model using the labeled seed dataset. We typically use

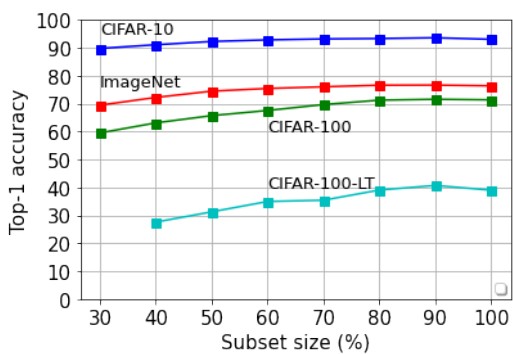

Figure 1: *We show the accuracy of ResNet56 models trained on subsets of different sizes, selected using our method on CIFAR-10, CIFAR-100, ImageNet, and CIFAR-100-LT. Models trained using subsets with 30-40% less data, achieve similar accuracy to the ones trained using the full dataset.*

a randomly selected 10% subset of the full dataset as seed. After this, as done in the standard batch-mode active learning setup, our algorithm uses this initial seed model to select a subset of the full dataset for which the labels will be revealed. We then train a new model with the selected subset. As shown in Fig. 1, our subset selection methods can identify subsets with 30% to 40% less data, which achieve performance similar to what we could get training with the full annotated dataset.

Most existing methods for data subset selection use some notion of *margin*, *representativeness*, or *diversity*. For example, the classical margin-based sampling selects images for which the prediction of the class labels are most uncertain. These could, for example, be the images having smallest difference between the highest and the second highest label probabilities (Lewis & Gale, 1994). Clustering-based objectives such as k-center (Sener & Savarese, 2018) or k-medoids (Kaufman & Rousseeuw, 1987) select cluster centers with rich representativeness to model the entire dataset. Diverse subsets are identified by casting the problem as the maximization of submodular functions (Wei et al., 2015), which are discrete analogues of convex functions.

We see two main limitations with existing methods. First, there is no single subset selection criterion that achieves the best performance on different classification datasets. For example we observe in our experiments (Sec. 6) that in certain datasets, diversity works better than margin, and vice versa. The second limitation is particularly relevant to margin-based methods where the selection is driven by the closeness of an image to a decision boundary. In a classification problem with $L$ labels, we have $\binom{L}{2}$ possible decision boundaries. Most margin-based methods greedily select images close to decision boundaries. This often results in a selection of too many images representing a small number of classes and boundaries, while ignoring many others.

To address the first limitation, we develop a unified algorithm based on maximization of a submodular function, which can combine different selection objectives. Importantly, we do this without sacrificing constant approximation guarantees. To address the second limitation, we incorporate class-balancing and boundary-balancing constraints in our optimization using intersection of matroids. Typical diversity promoting objective functions use an underlying nearest neighbor graph that models the distances between the training samples. In this work, We present an improved diversity objective function that models the simultaneous interaction of three nearby samples, while the standard approach typically considers the interaction of two nearby samples.

To summarize, our contributions are: (1) We develop a unified algorithm to combine different objective functions along with class-balancing and boundary-balancing constraints using the intersection of matroids. (2) We propose an improved diversity objective function based on 3-clique interactions in the nearest neighbor graph. (3) We outperform baseline methods on three standard classification datasets (CIFAR-10, CIFAR-100, and ImageNet) and a long-tailed classification dataset (CIFAR-100-LT).

## 2 RELATED WORK

**Data summarization and submodularity:** Submodularity (Nemhauser et al., 1978), often seen as the discrete analogue of convexity, can be used to model subset selection objectives associated with diversity and representativeness. While subset selection problems are mostly NP-hard, submodular optimization algorithms are efficient and come with constant approximation guarantees (Carbonell & Goldstein, 1998; Simon et al., 2007; Krause & Golovin; Golovin & Krause, 2010; Wei et al., 2015; Lin & Bilmes, 2011; Kaushal et al., 2019; Kim et al., 2016; Prasad et al., 2014; Golovin & Krause, 2010; 2011; Joseph et al., 2019; Zhou & Spanos, 2016; Qian et al., 2017; Das & Kempe, 2018; Elhamifar, 2019; Kothawade et al., 2021; Kaushal et al., 2021).

A submodular function that promotes representativeness and diversity was utilized in document summarization (Lin & Bilmes, 2011). Discriminant point process (DPP) are probabilistic models capable of selecting diverse subsets. They were used for minibatch selection in Mariet et al. (2020), and the maximum a posteriori (MAP) inference for DPP is submodular (Chen et al., 2018).

**Uncertainty sampling and active learning:** Active learning methods primarily focus on reducing the label annotation costs by selecting images to label that are more likely to yield the best model (See Settles (2009) for a detailed survey). Theoretical guarantees are usually about how quickly the version space, that maintains a set of good candidate classifiers, shrinks based on the availability of new data (Dasgupta et al., 2008). Uncertainty sampling selects the most challenging or uncertain

images first, in the hope that this will eliminate the need to label other, easier ones. In a classification setting, predictions from the initial model can provide class probabilities that can help identify images with high uncertainty. Possibilities include simple uncertainty measures from the best and second best class probabilities (Lewis & Gale, 1994; Scheffer et al., 2001), entropy measure (Holub et al., 2008; Joshi et al., 2009), and geometric distance to the decision boundary (Tong & Koller, 2002; Brinker, 2003), selection via proxies (Coleman et al., 2020), and query by committee (Gilad-Bachrach et al., 2005; Seung et al., 1992), where a committee of models are used to identify images where they most disagree.

**Coresets:** A coreset is a small set of points that approximates the shape of a larger point set. In an ML setting it is a small subset of images from a larger dataset such that the model trained on the coreset is competitive with the one learned from the entire dataset. It is shown in (Sener & Savarese, 2018; Wolf, 2011) that the average loss over any given subset of the dataset and the remaining points can be bounded, and that the minimization of this bound can be mapped to the k-center problem.

**Clustering:** Other classical techniques for subset selection include clustering algorithms such as k-medoids (Kaufman & Rousseeuw, 1987; Park & Jun, 2009), which minimizes the sum of dissimilarities between images belonging to a cluster and a point designated as its center. Agglomerative clustering was used to select subsets in Birodkar et al. (2019), where it was shown that at least 10% of images are redundant in ImageNet and CIFAR-10 training. In this work, we show that at least 30-40% of the images in CIFAR-10, CIFAR-100 and ImageNet are redundant. In Ash et al. (2020), k-MEANS++ algorithm is used to find a set of diverse and uncertain samples using the gradients of the final layer of the network.

In the context of interpretability, *prototypes* are subsets that best represent the entire dataset where a learned model performs the best, and *criticisms* are the images where a learned model does not do well. Both prototypes and criticisms can be modeled using maximum-mean discrepancy (MMD), that measures the difference between two distributions. The resulting optimization to compute the subsets can be cast as submodular maximization (Kim et al., 2016). Minimization of the MMD between the entire dataset and the selected subset can be seen as empirical risk minimization, and such an approach was used for batch mode active learning (Wang & Ye, 2015). Other classical techniques for prototype selection includes k-medoids (Bien & Tibshirani, 2011) clustering methods.

## 3 PROBLEM STATEMENT

We consider a classification setting with $L$ classes and $n$ training images $\mathcal{U} = \{x_1, \ldots, x_n\}$. Let $G = \{1, \ldots, n\}$ denote the indices in the dataset. Let $\psi : \mathcal{X} \to \mathbb{R}^L$ be a (trained) neural network that maps an input image to a probability distribution over the $L$ classes. The model $\psi$ can be seen as the composition of an *embedding function* $\phi : \mathcal{X} \to \mathbb{R}^D$ that maps an input to an $D$-dimensional embedding, and a *discriminator function* $h : \mathbb{R}^D \to \mathbb{R}^L$ that maps the embedding to output probability predictions.

**Subset selection and model evaluation.** We assume that we are given a small, annotated subset of $\mathcal{U}$, which we refer to as the *seed dataset* (this is 10% of $\mathcal{U}$ in our experiments). We use this seed dataset to train an initial model $\psi^{\text{seed}}$. Our goal is to use this initial model to select a subset $S$ of $G$, subject to a given annotation budget, such that when the images in $S$ are annotated we can get the best classification performance training a new model with them. To this end, using the seed model, we generate embeddings and predictions for all the input images in $\mathcal{U}$. Next, we construct a neighborhood graph $(G, E)$ where $E$ denotes the weighted set of edges associated with nearby training images in the embedding space $\mathbb{R}^D$. Our subset selection algorithm only depends on these embeddings, predictions, and the nearest-neighbor graph $(G, E)$. As in the standard active learning settings, we do not use the groundtruth labels in $\mathcal{U}$ while selecting subsets.

For modeling subset selection, we employ a submodular set function $f : 2^G \to \mathbb{R}$ that provides a utility value for every subset of $G$ using notions of uncertainty and diversity. We show that the class and boundary balancing constraints can be enforced using intersections of partition matroids. With this modeling, subset selection is essentially the task of finding the subset with maximum utility value while satisfying the matroid constraints. This inference problem is combinatorial in the number of training samples, and the brute-force solution is infeasible. For efficient inference, we rely on the submodular property of the objective function and the use of matroids for modeling balancing

constraints. Such an approach allows us to use an efficient greedy algorithm to maximize the objective function while satisfying matroid constraints with constant approximation guarantees.

Once the subset $S$ is computed, we request the groundtruth labels for the images in $S$. We use the annotated subset $\mathcal{U}_S \subseteq \mathcal{U}$ to train a model $\psi^S$ and evaluate its performance on a common held-out test set.

## 4 BACKGROUND

In Sec. 5 we show that our subset selection objective is both submodular and monotonic. We formally define these properties of a set function below.

**Definition 1.** *Let $\Omega$ be a finite set. A set function $F : 2^\Omega \to \mathbb{R}$ is **submodular** if for all $A, B \subseteq \Omega$ with $B \subseteq A$ and $e \in \Omega \setminus A$, we have $F(A \cup \{e\}) - F(A) \leq F(B \cup \{e\}) - F(B)$.*

This property is also referred to as diminishing returns since the gain diminishes as we add elements Nemhauser et al. (1978).

**Definition 2.** *A set function $F$ is **monotonically non-decreasing** if $F(B) \leq F(A)$ when $B \subseteq A$.*

In Sec. 5.1 we show how to incorporate class balancing and boundary balancing priors in the subset selection algorithm. We use algebraic structures called matroids and their intersections. We formally define them below.

**Definition 3.** *A **matroid** is an ordered pair $M = (\Omega, \mathcal{I})$ consisting of a finite set $\Omega$ and a collection $\mathcal{I}$ of subsets of $\Omega$ satisfying three conditions. (1) The collection contains the empty set, i.e., $\emptyset \in \mathcal{I}$. (2) If $I \in \mathcal{I}$ and $I' \subseteq I$, then $I' \in \mathcal{I}$. (3) If $I_1$ and $I_2$ are in $\mathcal{I}$ and $|I_1| < |I_2|$, then there is an element $e$ in $I_2 \setminus I_1$ such that $I_1 \cup \{e\} \in \mathcal{I}$.*

The members of $\mathcal{I}$ are called the independent sets of $M$. See Oxley (1992) for other, equivalent definitions of matroids.

**Definition 4.** *Let $\Omega_1, \ldots, \Omega_n$ be a partition of the finite set $\Omega$ and let $k_1, \ldots, k_n$ be positive integers. The ordered pair $M = (\Omega, \mathcal{I})$ is a **partition matroid** if $\mathcal{I} = \{I : I \subseteq \Omega, \quad |I \cap \Omega_i| \leq k_i, 1 \leq i \leq n\}$*

**Definition 5.** *Given two matroids $M_1 = (\Omega, \mathcal{I}_1)$ and $M_2 = (\Omega, \mathcal{I}_2)$ over the ground set $\Omega$, the **intersection of the matroids** is given by $M_1 \cap M_2 = (\Omega, \mathcal{I}_1 \cap \mathcal{I}_2)$.*

## 5 SUBSET SELECTION

We propose a unified formulation for uncertainty and diversity using set functions of the form $f : 2^G \to \mathbb{R}$, where $2^G$ is the power set of the vertex set $G$. We show that our unified objective function is submodular and montonic, and that the balancing constraints could be incorporated as matroids (formally defined in Sec. 4). This implies that our efficient greedy algorithm comes with constant approximation guarantees (Nemhauser et al., 1978).

We cast subset selection using a criterion $f$, as maximizing a corresponding set function under a budget constraint:

$$S^* = \arg\max_{S \subseteq G} f(S) \quad \text{s.t. } |S| = k. \tag{1}$$

Here $S^*$ is the optimal subset based on the selection criterion, and $k$ is the budget. Our functions consist of a sum of terms, and the individual terms are modeled using the distances between embeddings. These embeddings are produced by the initial model $\psi^{\text{seed}}$, which we train using the annotated seed dataset.

Our unified objective function is written as the conic combination of several individual functions based on uncertainty ($f_{\text{uncertainty}}$), diversity ($f_{\text{diversity}}$), and clique function ($f_{\text{triple}}$):

$$f(S) \quad = \quad \lambda_1 f_{\text{uncertainty}}(S) + \lambda_2 f_{\text{diversity}}(S) + \lambda_3 f_{\text{triple}}(S), \tag{2}$$

where $\lambda_i \geq 0, i \in \{1, \ldots, 3\}$. Each of these constituent functions have simple geometric interpretations (See Fig. 2), and we provide their analytical expressions below:

**Uncertainty based sampling.** We define the uncertainty-based utility function $f_{\text{uncertainty}}$ as:

$$f_{\text{uncertainty}}(S) = \sum_{i \in S} u(i), \qquad (3)$$

where $u(i)$ is a utility value for each example $i \in G$, based on its uncertainty. There are three popular choices for the utility functions using uncertainty: least confident, margin (Scheffer et al., 2001), and entropy. In this paper, we use margin, which is applicable in a multi-class setting, and gives preference to "hard to classify" or "low margin" examples (see Scheffer et al. (2001) for more details). More specifically:

$$u(i) = 1 - \Big( P(Y = bb|x_i) - P(Y = sb|x_i) \Big), \qquad (4)$$

where $bb$ and $sb$ denote the best and the second best predicted class labels for $x_i$ according to our initial model $\psi^{\text{seed}}$. $P(Y = \cdot|x_i)$ denotes the class probabilities predicted by the same model for $x_i$.

**Diversity.** Given a budget on the size of the subset, the *diversity* criterion promotes selection of images that are diverse or spread apart in the embedding space. We define a set function $f_{\text{diversity}}$ that evaluates diversity as follows:

$$f_{\text{diversity}}(S) = \sum_{i \in S} \text{unary}(i) - \gamma \sum_{i,j:\ (i,j)\in E,\ i,j\in S} s(i,j). \qquad (5)$$

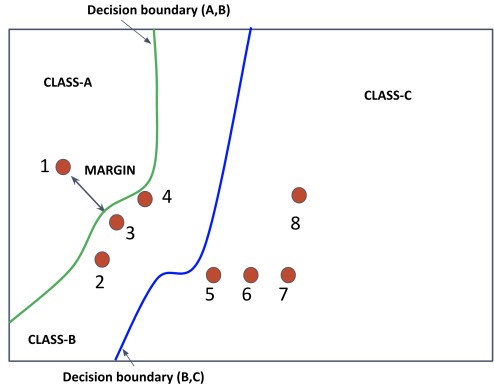

Figure 2: *A schematic with three classes (A,B,C), 8 training samples, and two decision boundaries for the label pairs (A,B) and (B,C). $f_{diversity}$ promotes selecting samples that are more spreadout and thereby gives higher utility for $\{1,3,7\}$ over $\{2,3,4\}$. $f_{uncertainty}$ prefers samples with high uncertainty or the ones closer to the decision boundary, and thus gives higher utility for $\{2,3,4\}$ over $\{1,3,7\}$. $f_{triple}$ avoids selecting triplets with negligible or small area, and thereby gives very low utility for subsets such as $\{5,6,7\}$ where the samples all lie on a straight line. The class balancing constraint would prefer a subset $\{1,3,7\}$, with samples from three classes, to a single class set $\{2,3,4\}$. The boundary-balancing constraint would prefer $\{4,5\}$, which spans over two decision boundaries, to $\{3,4\}$ lying near a single decision boundary.*

The unary terms $\text{unary}(i)$ capture the utility of an individual image and the pairwise terms $s(i,j)$ are the similarity between a pair of images from the selected subset that have an edge $(i,j) \in E$ in the nearest neighbor graph $(G, E)$. As discussed in the Sec. 6, we use the cosine similarity in the embedding space as the similarity measure.

The hyperparameter $\gamma \geq 0$ determines the relative importance of the unary and pairwise terms. The pairwise term limits redundancy. If two points $i$ and $j$ with high similarity are both selected, we incur a penalty based on the similarity between them.

To ensure monotonicity of $f_{\text{diversity}}$, we can use a constant unary term, $\text{unary}(i) = \max_{l \in G} \sum_{j:(l,j) \in E} s(l,j)$.

**Lemma 1.** *The function $f_{diversity}$ is monotonically non-decreasing and submodular.*

The proof is available in the Appendix.

**Clique function.** Given a nearest neighbor graph, we identify cliques of size three by checking for edges among the k-nearest neighbors of every sample. Given a set of cliques given by $\mathcal{T}$, we define $f_{\text{triple}}$ as follows:

$$f_{\text{triple}}(S) = \sum_{i \in S} \alpha(i) - \eta \sum_{i,j,k:\ (i,j,k)\in\mathcal{T},\ i,j,k\in S} t(i,j,k). \qquad (6)$$

where $\alpha(i)$ denotes the number of triple cliques involving the sample $i$, and $t(i,j,k)$ is 1 when the volume consumed by the embeddings of the triplet $\{i, j, k\}$ is less than a threshold $\mathcal{T}_{thresh}$, and 0 otherwise. The volume computation in higher dimensional space involves outer product (Lundholm & Svensson, 2009), similar to cross-product in 3-dimensional space. For computational efficiency, we approximate the volume using the area of triangle based on the three side lengths obtained from the nearest neighbor graph. We can extend the idea of triple cliques to higher orders (Kolmogorov &

Zabin, 2004; Kohli et al., 2007; Ramalingam et al., 2017), but this can be computationally expensive when we consider millions of training samples.

We prove the following lemma in the Appendix.

**Lemma 2.** *The function $f_{triple}$ is monotonically non-decreasing and submodular.*

We note that there are other ways to maximize the diversity function. For example, by replacing the similarity term $s(i, j)$ with a metric distance function, the diversity criteria can be seen as the max-sum k diversification problem, for which algorithms exist with constant approximation guarantees (Borodin et al., 2017).

Any conic combination of submodular functions is submodular (Nemhauser et al., 1978), and thus $f(S)$ is submodular.

### 5.1 BALANCING CONSTRAINTS

In this section we show how we incorporate balancing constraints in our unified formulation.

**Class balancing constraint.** Subsets selected based on diversity or margin criteria can lead to severe class imbalance. Consequently, models trained using these imbalanced subsets can show superior performance on common classes, and inferior performance on rare classes Cui et al. (2019). A reweighted loss function can partially address this problem but performance on classes with no, or very few representatives will still suffer.

To overcome this problem we incorporate explicit class-balancing constraints in our formulation. We do this using a partition matroid constraint, a choice that allows us to maintain our approximation guarantees. Consider a partition $\{G_1, G_2, \ldots, G_L\}$ of the full dataset $G$. Here $L$ denotes the number of classes and $G_i$ consists of the images belonging to class $i$, where pseudo-labels are computed using the initial (seed) model. Let $k_1, \ldots, k_L$ be user-defined positive integers specifying the sizes of the desired subsets. We choose subsets that are independent sets of the partition matroid $M = (G, \mathcal{I}_p)$:

$$\mathcal{I}_c = \{I : I \subseteq G, \quad |I \cap G_i| \leq k_i, 1 \leq i \leq L\}. \tag{7}$$

**Boundary balancing constraint.** The initial model $\psi^{\mathcal{I}}$ that we trained using the seed dataset, defines a set of decision boundaries for every pair of classes. Formally, the decision boundary for a pair of classes $\{a, b\}$ is given by:

$$\mathcal{D}_{\{a,b\}} \triangleq \{x; P(Y = a|x) = P(Y = b|x)\}, \tag{8}$$

where $P(y|x)$ are the posterior probabilities from the seed model. In other words, these are the set of points where the posterior class probabilities for two different classes agree. Decision boundaries can be defined in input space, output space, or in the space defined by the intermediate layers in the network Elsayed et al. (2018). Since we are only interested in associating an image to its closest decision boundary, our formulation does not require computation of the exact distance to the boundary. Instead we simply use predicted class probabilities.

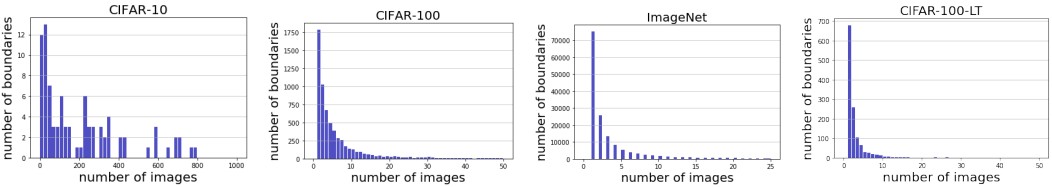

Figure 3: *Histogram of the number of images associated with the class boundaries in CIFAR-10, CIFAR-100, ImageNet, and CIFAR-100-LT datasets.*

We define the *margin score* of an image $x_i$ (given predictions from the initial model) as

$$m(x_i) = 1 - \Big( P(Y = bb|x_i) - P(Y = sb|x_i) \Big), \tag{9}$$

where $bb$ and $sb$ correspond to best and the second best class labels, respectively. We associate $x_i$ with the boundary $(bb, sb)$ if $m(x) > \tau$, for some threshold $\tau$.

We observe that the number of class boundaries grows quadratically with the number of classes and conjecture that many datasets with a large number of classes do not contain any images close to many decision boundaries. Our goal is to design a subset selection algorithm that promotes selection of images near as many decision boundaries as possible.

To quantify this observation, we use predictions from our initial model for ImageNet to associate any image in the dataset which has a margin score greater than $\tau = 0.05$ to the corresponding class boundary. Since ImageNet has 1000 classes, we have $\binom{1000}{2}$ boundaries. We found that 15% of all possible boundaries were covered.

Let $\mathcal{D} = \{b(x); x \in \mathcal{U}\}$ denote the set of all the decision boundaries associated with all the images in the training set $\mathcal{U}$, and $b(x)$ is a function that associates a given image to a certain decision boundary or to the empty set $\emptyset$, based on the margin score. In Fig. 3, we show the histogram of the number of images associated with different decision boundaries. We note that it is possible to associate an image to multiple decision boundaries, and defer exploring this option to future work.

Similar to the class-balancing problem, we consider a partition $\{G_{b_1}, G_{b_2}, \ldots, G_{b_M}\}$ of the full dataset $G$, where $G_{b_i}$ denotes the set of images belonging to the decision boundary $b_i \in \mathcal{D}$.

Let $d_1, \ldots, d_M$ be user-defined positive integers specifying the maximum number of images near different boundaries. Let us consider the following partition matroid $M = (G, \mathcal{I}_d)$:

$$\mathcal{I}_d = \{I : I \subseteq G, \quad |I \cap G_{b_i}| \le d_i, 1 \le i \le M\} \tag{10}$$

In order to impose both class-balancing and boundary-balancing constraints, our selected subset should be the independent set of both the partition matroids given by $(G, \mathcal{I}_c)$ and $(G, \mathcal{I}_d)$.

Note that the intersection of two matroids is not a matroid in general. However, Fisher et al. (1978) shows that applying the greedy algorithm to a submodular monotonic objective with matroid intersection constraints maintains approximation guarantees.

## 5.2 GREEDY ALGORITHM

Subset selection using class- and boundary-balancing constraints is given by:

$$S^* = \arg\max_{S \subseteq G} f(S) \quad \text{s.t.} \ |S| = k, S \in \mathcal{I}_c, S \in \mathcal{I}_d, \tag{11}$$

where $f$ is our unified objective. The greedy algorithm for selecting a subset of size $k$ is given in Algorithm 1.

---

**Algorithm 1:** Greedy algorithm

---

1. Initialize $S = \emptyset$;
2. Let $s = \arg\max_{s' \in G} f(S \cup \{s'\}) - f(S)$ such that $S \cup \{s'\} \in \mathcal{I}_c \cap \mathcal{I}_d$;
3. If $s \ne \emptyset$ and $|S| < k$ then $S = S \cup \{s\}$, go to 2.;
4. $S$ is the required solution;

---

Given that $f$ is monotonic and submodular, a theorem from Fisher et al. (1978) implies that we have a constant approximation guarantee for the greedy algorithm according to the bound $f(S_{greedy}) \ge \frac{1}{p+1} f(S_{OPT})$, where $f(\emptyset) = 0$ and $p$ is the number of matroid constraints. Since we have two matroids, we have $f(S_{greedy}) \ge \frac{1}{3} f(S_{OPT})$.

## 5.3 COMPUTATIONAL COMPLEXITY

The greedy algorithm can be efficiently implemented using a priority queue. Given a $k$-nearest graph on the embeddings and the uncertainty estimates, the proposed algorithm has a complexity of $O(|V| + nk^2 log^2(|V|))$, where $V$ is the set of all samples and $n$ is the size of the subset. Without the triple cliques its complexity is $O(|V| + nklog(|V|))$, which is the same as that of $k$-center. For implementation reference, we provide pseudo-code in Algorithm 2. In experiments, we choose $k = 10$, and the additional overhead from triple-clique constraint is minimal.

---

**Algorithm 2:** Pseudocode for Subset Selection.

---

1. **Input:** $k$-NN graph $G = (V, E)$.

2. Initialize an empty priority queue $pq$.

3. For all $v \in V$ do $pq \leftarrow \text{ADD}(v, w(v))$, where the weight $w(v)$ is based on unary such as uncertainty.

4. Initialize $S = \{c\}$, where $(c, w) = \text{POP}(pq)$.

5. While $|S| < k$ and NOT-EMPTY$(pq)$ do:

   - 6. $(c, w) \leftarrow \text{POP}(pq)$.

   - 7. If $S \cup c$ satisfies balancing constraints, $S \leftarrow S \cup c$.

   - 8. For all $n_1 \in \text{NEIGHBOR}(c)$ and $n_1 \notin S$, do $pq \leftarrow \text{UPDATE}(n_1, w(n_1) - \gamma s(n_1, c))$.

       - 9. For all $n_2 \in \text{NEIGHBOR}(c)$ and $n_2 \in S$, do
         $pq \leftarrow \text{UPDATE}(n_1, w(n_1) - \gamma s(n_1, c) - \eta t(c, n_1, n_2))$.

9. Output $S$.

---

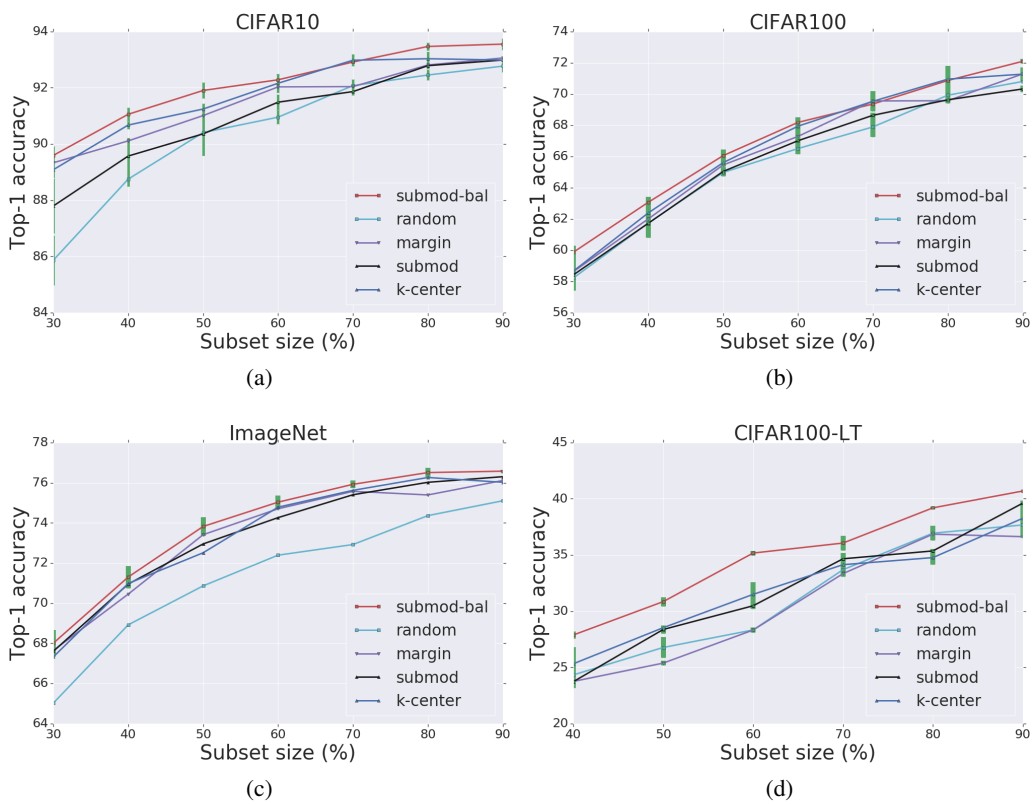

(a)

(b)

(c)

(d)

Figure 4: *All the Top-1 accuracy numbers are computed by averaging three trials. For the ImageNet, the error bars are only shown for our algorithm* SUBMOD-BAL *with one standard deviation.*

## 6 EXPERIMENTS

**Datasets.** We use four datasets: CIFAR10 (Krizhevsky et al., 2009), CIFAR100 (Krizhevsky et al., 2009), ImageNet (Russakovsky et al., 2014), and CIFAR100-LT (Menon et al., 2020). The last one is a long-tailed dataset where the head class is 100 times more frequent than the least-frequent class (Menon et al., 2020). We do not apply any long-tail techniques as we want to compare the performance of vanilla ERM across different methods.

**Initial model.** For each of the datasets we select a random seed subset of size $10\%$ and use it to train a ResNet-56 (He et al., 2016) initial model. We use the initial model to generate predictions and

embeddings for all the images in the dataset. The embeddings are $64$-dimensional for CIFAR and $2048$-dimensional for Imagenet. We construct the neighborhood graph $(G, E)$ by finding $k = 10$ nearest neighbors of each image in embedding space, using the fast similarity search of Guo et al. (2020). We evaluated different similarity measures such as cosine similarity, $L_1$, and $L_2$ distances, and found no significant differences. All the reported experiments use cosine similarities.

**Hyperparameters.** For the CIFAR10, CIFAR100, and CIFAR100-LT experiments we used 450 epochs and the learning rate is divided by 10 at the following epochs:15, 200, 300, and 400. The base learning rate was set to 1.0. For ImageNet we used 90 epochs and the learning rate is divided by 10 at the following epochs: 5, 30, 60, and 80. The base learning rate was set to 0.1. In all the datasets, we used SGD with momentum 0.9 (with nestrov) for training. When trained with the full training set, the top-1 accuracies for CIFAR-10, CIFAR-100, ImageNet, and CIFAR-100-LT are 93.04%,71.37%, 76.39%, and 39.01%, respectively.

**Evaluation.** We show four baselines: RANDOM, MARGIN, SUBMOD, and $k$-CENTER (Sener & Savarese, 2018). SUBMOD baseline is the function in Equation 2 without the triple clique term and the balancing constraints. The parameters $(k_1, \ldots, k_L)$ in the class-balancing partition matroid are set based on the desired subset size. In other words, $k_i$ sets an upper limit on the number of images from a specific class. In our experiments, we set this value to be $\frac{\eta n}{L}$, where $n$, $\eta$, and $L$ denote the number of images, fraction of the training set, and the number of labels, respectively. In the case of boundary balancing matroid, we set the parameters $d_i$ to be $\max(1, \eta n_i)$, where $n_i$ denotes the number of images associated with the decision boundary $b_i \in \mathcal{D}$. For every subset, we train three models and averaged performance is shown in Fig 4. For our algorithm SUBMOD-BAL we use $\lambda_1 = 0.7$, $\lambda_2 = 0.30$, and $\lambda_3 = 1$ in Equation 2. The baselines RANDOM and MARGIN do not have any hyperparameters. For $k$-CENTER and SUBMOD, we use the same nearest neighbor graph with the same number of neighbors.

We outperform other methods marginally in standard datasets and significantly in the long-tailed dataset. The $k$-CENTER (Sener & Savarese, 2018) performs close to the proposed method and outperforms standard submdoular algorithm. Note that on all these datasets, accuracies with either 80% or 90% subsets, or both, exceed that with the full dataset. This is not surprising since completely eliminating some examples in the training set has shown to be beneficial in prior work (Kubat & Matwin, 1997; Wallace et al., 2011).

**Ablation study:** The balancing constraints show an average improvement of 0.61% in CIFAR datasets, and 2.83% in CIFAR100-LT. The clique function shows an improvement of 0.49% in CIFAR and 1.13% in CIFAR100-LT dataset. We evaluated the role of $k$, the number of nearest neighbors in k-nearest neighbor graph construction. While we used $k = 10$ in all our experiments the marginal benefit in using $k = 100$ and $k = 1000$ are given by $+0.15\%$ and $+0.25\%$, respectively.

## 7 DISCUSSION

We propose a triple-clique diversity objective and balancing constraints on class labels and decision boundaries. We showed that the incorporation of balancing constraints usually leads to better subsets compared to standard baselines, and the impact is highly significant in training with long-tailed datasets such as CIFAR100-LT. We also show that some popular image classification datasets have at least 30-40% redundant images.

On standard datasets, the difference between our method and $k$-center is marginal, and this can be due to a deeper connection. The $k$-center algorithm can be reduced to a series of dominating set problems (Vazirani, 2001). Dominating set can be reduced to set cover, which is also a submodular problem. While $k$-center has connection to submodularity, our proposed framework does not generalize $k$-center. It will be beneficial to find a principled way to combine these methods.

## ETHICAL STATEMENT

Training large models are resource intensive and labeling data sets is also extremely labor-intensive, requiring significant human effort. For example, detailed semantic segmentation for a single image can take up to 60 minutes. In this work, we develop subset selection methods that can lead to accurate models with less computations and data. This will eventually lead to reduced human labeling efforts, and less utilization of computational resources.

## REPRODUCIBILITY STATEMENT

The basic algorithm to find the subset is a standard greedy algorithm and it is easy to implement. We also provided the pseudocode in Algorithm 2. This paper is only about subset computation and the training code is not changed. All the hyperparameters are listed in experiment section. We will release the code, the seed model, and the nearest neighbor graph for other researchers to compare their models.

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

## A   APPENDIX

**Lemma 1.** *The function $f_{diversity}$ is monotonically non-decreasing and submodular.*

*Proof.* The function $f_{\text{diversity}}$ is given by

$$f_{\text{diversity}}(S) = \sum_{i \in S} \max_{l \in G} \sum_{j:(l,j) \in E} s(l,j)$$
$$-\gamma \sum_{i,j:\ (i,j) \in E,\ \ i,j \in S} s(i,j). \tag{12}$$

We will show that the function is monotonically non-decreasing and submodular when $0 \leq \gamma \leq 1$.

We will first show that the function is monotonically non-decreasing. Consider an element $e \in G \backslash S$:

$$f_{\text{diversity}}(S \cup e) - f_{\text{diversity}}(S) = \max_{l \in G} \sum_{j:(l,j) \in E} s(l,j) -$$
$$\gamma \sum_{j:\ (e,j) \in E,\ \ j \in S} s(e,j) \tag{13}$$

In the first term, $j \in G$, whereas in the second $j \in S$. The first term is a maximum over all samples in $G$, including $l = e$. Since $\lambda \leq 1$ and $S \subseteq G$, we have $f_{\text{diversity}}(S \cup e) - f_{\text{diversity}}(S) \geq 0$ and thus the function in Eq. 12 is monotonically non-decreasing.

Next, we will show that the function $f_{\text{diversity}}(S)$ is submodular. In order to do this, we show the following when $A, B \subseteq G$, $B \subseteq A$, and $e \in G \backslash A$:

$$f_{\text{diversity}}(A \cup e) - f_{\text{diversity}}(A) \leq f_{\text{diversity}}(B \cup e) - f_{\text{diversity}}(B) \tag{14}$$

We expand the left hand side of the above expression using the definition of $f_{\text{diversity}}(S)$:

$$f_{\text{diversity}}(A \cup e) - f_{\text{diversity}}(A) = \max_{l \in G} \sum_{j:(l,j) \in E} s(l,j) -$$
$$\lambda \sum_{j:\ (e,j) \in E,\ \ j \in A} s(e,j) \tag{15}$$

We expand the right hand side as follows:

$$f_{\text{diversity}}(B \cup e) - f_{\text{diversity}}(B) = \max_{l \in G} \sum_{j:(l,j) \in E} s(l,j) -$$

$$\gamma \sum_{j:\ (e,j) \in E,\ \ j \in B} s(e,j) \tag{16}$$

Subtracting Eq. 16 from Eq. 15 we have:

$$(f_{\text{diversity}}(A \cup e) - f_{\text{diversity}}(A)) \quad -$$
$$(f_{\text{diversity}}(B \cup e) - f_{\text{diversity}}(B)) \quad =$$

$$\gamma \sum_{j:\ (e,j) \in E,\ \ j \in B} s(e,j) - \gamma \sum_{j:\ (e,j) \in E,\ \ j \in A} s(e,j) \tag{17}$$

Since $\gamma \geq 0$, $s(.,.) >= 0$, and $B \subseteq A$, we satisfy Eq. 14. $\qquad\square$

**Lemma 2.** *The function $f_{triple}$ is monotonically non-decreasing and submodular.*

*Proof.* The function $f_{\text{triple}}$ is given by

$$f_{\text{triple}}(S) = \sum_{i \in S} \alpha(i) - \eta \sum_{i,j,k:\ (i,j,k) \in \mathcal{T},\ \ i,j,k \in S} t(i,j,k). \tag{18}$$

We will show that the function is monotonically non-decreasing and submodular when $0 \leq \eta \leq 1$.

We will first show that the function is monotonically non-decreasing. Consider an element $e \in G \backslash S$:

$$f_{\text{triple}}(S \cup e) - f_{\text{triple}}(S) = \alpha(e) -$$

$$\eta \sum_{e,j,k:\ (e,j,k) \in \mathcal{T},\ \ e,j,k \in S} t(e,j,k) \tag{19}$$

The first term $\alpha(e)$ is the number of triple cliques associated with $e$, and the maximum value of the second term is $\alpha(e)$ since $0 \leq \eta \leq 1$. Thus we have $f_{\text{triple}}(S \cup e) - f_{\text{triple}}(S) \geq 0$ and thus the function in Eq. 18 is monotonically non-decreasing.

Next, we will show that the function $f_{\text{triple}}(S)$ is submodular. In order to do this, we show the following when $A, B \subseteq G$, $B \subseteq A$, and $e \in G \backslash A$:

$$f_{\text{triple}}(A \cup e) - f_{\text{triple}}(A) \leq f_{\text{triple}}(B \cup e) - f_{\text{triple}}(B) \tag{20}$$

We expand the left hand side of the above expression using the definition of $f_{\text{triple}}(S)$:

$$f_{\text{triple}}(A \cup e) - f_{\text{triple}}(A) = \alpha(e) -$$

$$\eta \sum_{e,j,k:\ (e,j,k) \in \mathcal{T},\ \ e,j,k \in A} t(e,j,k) \tag{21}$$

We expand the right hand side as follows:

$$f_{\text{triple}}(B \cup e) - f_{\text{triple}}(B) = \alpha(e) -$$

$$\eta \sum_{e,j,k:\ (e,j,k) \in \mathcal{T},\ \ e,j,k \in B} t(e,j,k) \tag{22}$$

Subtracting Eq. 22 from Eq. 21 we have:

$$(f_{\text{triple}}(A \cup e) - f_{\text{triple}}(A)) \quad -$$
$$(f_{\text{triple}}(B \cup e) - f_{\text{triple}}(B)) \quad =$$

$$\eta \sum_{e,j,k:\ (e,j,k) \in \mathcal{T},\ \ e,j,k \in B} t(e,j,k) - \tag{23}$$

$$\eta \sum_{e,j,k:\ (e,j,k) \in \mathcal{T},\ \ e,j,k \in A} t(e,j,k) \tag{24}$$

Since $\eta \geq 0$, $t(.,.,.) >= 0$, and $B \subseteq A$, we satisfy Eq. 20.

$\qquad\square$

