# OpenReview forum: "Less data is more: Selecting informative and diverse subsets with balancing constraints"
_ICLR.cc/2022/Conference — ICLR 2022 Submitted_

### Official Review · Reviewer_oEcZ · 2021-10-31

**Correctness:** 3
**Technical Novelty And Significance:** 2
**Empirical Novelty And Significance:** 2
**Recommendation:** 5
**Confidence:** 4

**Main Review:**

Strengths:
1.	The paper is easy to read and understand.
2.	The proposed boundary balancing constraint is interesting.
3.	It is good to do experiments on large datasets.

Weaknesses:
1.	The authors claimed that “we develop a unified algorithm based on maximization of a submodular function, which can combine different selection objectives” in Introduction. However, I can’t see the necessary or benefit of using “submodular function and matroid intersection” in this paper, except that some prior works about submodular show the guarantee of approximation to the original dataset in the some feature/output space. The question is that the subset with optimal approximation of a dataset in some feature space doesn’t guarantee that it is the optimal for training neural networks. It is known that random selection can also perfectly approximate the geometry of original dataset (Chen et al 2012) as long as the subset is large enough. In addition, it is also unknown that such approximation guarantee of using submodular function is better or faster to reach than other approximation methods like herding, k-center (Sener & Savarese, 2018), k-medoids etc.
2.	The authors admitted that “the difference between our method and k-center is marginal” in Discussion. The performance comparison also verified it. The performance improvement of the proposed method is mainly due to balancing constraints. More detailed ablation study on each individual balancing constraint should be given. Otherwise, it is unclear which constraint works.
3.	The experiments lack the comparison to state-of-the-art subset selection methods.
4.	In the balancing constraints, the image number of each class is pre-defined and the underlying assumption is that the class number is known. Is this assumption realistic in the experimental setting (active learning)?

(Chen et al 2012) Super-Samples from Kernel Herding


**Summary Of The Paper:**

This paper proposed a new subset selection method with novel diversity objective and balancing constraints. The authors prove that the proposed sample selection criterion is a submodular function. Hence its greedy algorithm comes with approximation guarantees (Nemhauser et al., 1978). The experiments on popular image classification datasets show that the proposed method is slightly better than k-center (Sener & Savarese, 2018).

**Summary Of The Review:**

The proposed method achieves slight improvement over a classic method (k-center) by introducing new objective and constraints. The paper lacks deep analysis and solid experiments that explain why the new components are better. The contribution is somewhat incremental.

---

> ### Author Response · Authors · 2021-11-23
> **Response to Reviewer oEcZ**
>
> Thanks for the detailed feedback. We address the major concerns below.
>
> **Guarantees and computational complexity**
>
> Great suggestion! Yes, there is no guarantee that approximation of the original dataset in some feature/output space is optimal for training deep neural networks. The approximation guarantees are all related to the objective function we use for tackling different selection criteria. We also completely agree that random subset can also perfectly capture diversity in large datasets (Chen et al. 2012). We will add a reference to Chen et al. 2012 and discuss this. While a random subset can indeed capture diversity in most of the cases, the use of submodular or k-center can ensure that the inter distance between different samples satisfy certain constraints. In practice, this does provide some lift but not a significant one in all cases.
>
> Unfortunately, there is a big gap between theory and practice in active learning methods applicable to deep learning. Theoretical results are usually about how to achieve a sufficiently accurate model with a bound on the number of queries used.
>
> While the main contribution of this paper is in the optimality guarantees and the formulation of the constraints, the submodular objective function is closely related to the k-center and classical uncertainty methods that have been theoretically studied in active learning settings.
>
> A recent work on k-center [Sener et al. 2017] shows that the minimization of the target risk corresponds to solving a k-center objective. However, note that this does not correspond to an objective function involving both diversity and margin. In the case of linear separators, margin sampling can use O(d log(1/epsilon)) labeled examples to get a model with error less the epsilon when the data lies in a ball of $R^d$ [ Balkan et al. 2007].
>
> **Computational complexity and relation to k-medoids, k-center, and kernel herding**
>
> K-medoid, K-center and the proposed algorithm (in the absence of triple cliques) can all be implemented using priority queue and the complexity is O(|V |log(|V |) + nklog(|V |)) where n is the size of the subset, V is the set of all samples, and k is the number of neighbors. In the presence of triple cliques, the proposed algorithm has a complexity of O(|V |log(|V |) + nk^2log2 (|V |)), and this is almost the same as the earlier for constant small values of k=10.
>
> Kernel herding also uses a greedy algorithm to explore the reproducing kernel Hilbert space (RKHS). The goal is to approximate an original point set with another one such that the kernel density estimates match. The kernel herding method adds one sample at each step that most decreases an error metric. The complexity of the kernel herding [Chen et al. 2012] algorithm is O(|V|n), which is larger than the complexity of the proposed algorithm.
>
> **Difference with our method and k-center method**
>
> Our method performs similar to k-center due to a deeper mathematical connection and for convenience, we mention it again here. The k-center algorithm can be reduced to a series of dominating set problems [61]. Dominating set can be reduced to set cover, which is also a submodular problem.
>
> The balancing constraints show an average improvement of 0.61% in CIFAR datasets, and 2.83% in CIFAR100-LT. One can also use the balancing constraints individually for classes and boundary constraints. At a high level, class-level balancing constraint only looks at associating one class for every image, and boundary-level balancing constraint looks at associating two classes to every image. Please note that our formulation uses the intersection of matroid constraints for addressing class and boundary balancing constraints. We observed that class-level boundary constraints provided more improvement (2.13%) compared to boundary-level constraints (1.69%) on CIFAR100-LT.
>
> **Additional baselines**
>
> We compared our work to k-center, margin, submodular, and random baselines. We did compare our proposed method with k-medoids (CIFAR and ImageNet) and BADGE (CIFAR) and they are similar or inferior to margin and k-center methods. Please note the margin is a classical algorithm, but it is still the gold standard and not an easy baseline for even newer methods. This was also observed in a recent work [1]. In the final version, we can add other baselines such as k-medoids and BADGE [1].
>
> **Class and boundary statistics in active learning settings**
>
> Insightful remark. Yes, in all our experiments, we only use predicted labels and we don’t rely on the actual number of labels or decision boundaries with respect to the true labels. So, this does not violate any active learning principle. In some sense, we are working with pseudo-labels as in the case of semi-supervised learning methods, but we still observe improvement in long-tailed datasets. Other reviewers such as Cp2k commented on this as well.

---

> > ### Comment · Reviewer_oEcZ · 2021-11-28
> > **Thanks for authors' response**
> >
> > Thanks for the response! Most of my questions are answered in text. The submission can be improved by adding these comparison and analysis. However, I still don't recommend the acceptance, as I am not convinced by its practical contribution or impact that can obviously advance coreset selection research.

---

### Official Review · Reviewer_yngU · 2021-11-01

**Correctness:** 2
**Technical Novelty And Significance:** 2
**Empirical Novelty And Significance:** 2
**Recommendation:** 3
**Confidence:** 4

**Main Review:**

## Pros
- Formulation of the optimization problem is well motivated.
- The proposed algorithm is clearly described using rigorous equations.
- Proofs of important properties of the method (submodularity of objective functions) are provided in the appendix.

## Cons
- The motivation for the triplet clique objective (6) is unclear. The function appears to encourage diversity of subset $S$ as (5) has addressed, and no discussion (theoretical or empirical) is provided as to whether the advantage of including (6) can offset the additional computational complexity it introduces.
- Balancing constraints seem to play the most important role from the empirical results. This puts into question the importance of the first claimed contribution of the paper, namely the unified objective function of (2), as `submod` without balancing does not significantly outperform other baselines. However, no ablation experiments are included to study the importance of boundary-level balancing constraints in addition to class-level balancing.
- Theoretical analysis or ablation studies for important hyperparameters of the proposed method are also missing from the experiments. Some examples for these include the loss weights $\lambda_1, \lambda_2$ and $\lambda_3$; the choice of balancing hyperparameters $k_{1:L}$, $d_{1:M}$; the margin score threshold $\tau$.
- Lack of comparison to state-of-the-art active learning methods. While the paper compared to core-set selection, stronger baselines (e.g. [A-C]) were not included. In addition, comparisons in the paper are limited to classification datasets, but not for more label-intensive tasks such as semantic segmentation, which [B, C] has studied.
- Error bars for methods other than `submod-bal` were not reported. This makes it hard to determine the statistical significance of results when comparing two approaches.

## Minor comments
- Equation (9) seems redundant given the margin score has been defined in (4). I wonder if it makes more sense to instead define the boundary assignment function $b(x_i) = (bb(x_i), sb(x_i))$?

[A] Yoo, Donggeun, and In So Kweon. "Learning loss for active learning." *Proceedings of the IEEE/CVF Conference on Computer Vision and Pattern Recognition*. 2019.

[B] Sinha, Samarth, Sayna Ebrahimi, and Trevor Darrell. "Variational adversarial active learning." *Proceedings of the IEEE/CVF International Conference on Computer Vision*. 2019.

[C] Zhang, Beichen, et al. "State-relabeling adversarial active learning." *Proceedings of the IEEE/CVF Conference on Computer Vision and Pattern Recognition*. 2020.

**Summary Of The Paper:**

This paper presents a new method for batch active learning in deep neural networks, which aims to progressively construct compact subsets from a training dataset that maximize accuracy of learned models. The authors propose to address this problem by optimizing a submodular set function, expressed as a weighted combination of uncertainty, diversity, and triplet objectives, under a set of balancing constraints on classes and decision boundaries. The resulting optimization problem is solved using a greedy algorithm. Experiments are conducted on CIFAR10, CIFAR100, ImageNet and CIFAR100-LT datasets, where the proposed approach outperforms all baselines, with larger improvements under a class-imbalanced setting (CIFAR100-LT).

**Summary Of The Review:**

I appreciate the effort of authors towards a unified framework for sample selection in active learning. However, as detailed in the Cons section above, the importance of many elements of the proposed method are not adequately justified through theoretical or empirical results, and comparisons to prior work are lacking in number of baselines as well as range of tasks. I believe additional experiments/analysis addressing these concerns are needed to support the major claims of the paper.

---

> ### Author Response · Authors · 2021-11-23
> **Response to Reviewer yngU**
>
> Thanks for the detailed feedback. We address the major concerns below.
>
> **Need better justification and discussion about $f_{triple}$**
>
> Thanks for this clarification. Reviewer 4ATq also had a question related to the role of triple cliques. For convenience, we share the response here.The basic idea of the triple clique is straightforward and inspired from many vision papers. We can consider the original submodular objective function as a Boolean function with only pairwise terms as shown below:
>
> $f(x_1, …, x_n) = \sum_{i} a_i x_i + \sum_{i,j} b_{ij} x_i x_j + \sum_{ijk} c_{ijk} x_i x_j x_k$
>
> If the coefficients $a_i$, $b_{ij}$, and $c_{ijk}$ satisfy certain conditions the overall objective function is submodular. Mathematically, we can not find a function that uses only pairwise terms to approximate a function involving triple clique terms. Let us consider a simple example below:
>
> $f(x_1, x_2, x_3) = -x_1x_2x_3$
>
> We can have functions such as $g(x_1,x_2,x_3) = - \frac{1}{3}(x_1 + x_2 + x_3)$ that can match the triple clique function when $(x_1=0,x_2=0,x_3=0)$ and $(x_1=1,x_2=1,x_3=1)$, but not at for all combinations. While minimizing submodular functions, there are techniques that use additional variables to minimize the functions [Kolmogorov et al. PAMI 2004] by casting them as pairwise functions, but this is not generally not an easy thing to do.
>
> Regarding complexity, the proposed algorithm can be implemented using priority queue and the complexity is O(|V |log(|V |) + nklog(|V |)) without triple cliques, where n is the size of the subset, V is the set of all samples, and k is the number of neighbors. In the presence of triple cliques, the proposed algorithm has a complexity of O(|V |log(|V |) + nk^2log2 (|V |)), and this is almost the same as the earlier for constant small values of k=10.
>
> **Balancing constraints**
> Reviewers oEcZ and Cp2k also shared the same question. The balancing constraints show an average improvement of 0.61% in CIFAR datasets, and 2.83% in CIFAR100-LT. One can also use the balancing constraints individually for classes and boundary constraints. At a high level, class-level balancing constraint only looks at associating one class for every image, and boundary-level balancing constraint looks at associating two classes to every image. Please note that our formulation uses the intersection of matroid constraints for addressing class and boundary balancing constraints. We observed that class-level boundary constraints provided more improvement (2.13%) compared to boundary-level constraints (1.69%) on CIFAR100-LT.
>
> **Theoretical analysis or ablation studies for important hyperparameters**
>
> Note that there are some constraints on lambdas to ensure submodularity. As we mentioned, we have already provided the hyperparameters used in the paper e λ1 = 0.7, λ2 = 0.30, and λ3 = 1 and the other hyperparameters are dataset dependent and we have provided the expressions in the paper. Reviewer 4ATq had the same question. Our uncertainty values are in the range [0, 1] and the cosine similarity values vary from -1 to 1. We used $k=10$. We set a larger value for the coefficient of margin compared to diversity since we have $k=10$. T_{thresh} is chosen to just provide some additional benefit from the triple clique without an aggressive search. We did not perform an elaborate hyperparameter search, but we just tested around 10 or fewer hyperparameter settings manually based on the relative importance of different criteria.
>
> Thanks for pointing to papers [A-C]. We will cite them and add a discussion to these papers. Yes, indeed semantic segmentation and other label-intensive tasks would be more appropriate for active learning work. While the diversity and triple-clique functions may apply for semantic segmentation, the balancing constraints can not be extended directly to semantic segmentation tasks and this may require newer approaches. In the spirit of active learning to reduce human label annotation tasks, semantic segmentation would be a great problem for us to study in the future.
>
> **Error bars on all datasets**
>
> Thanks for this suggestion. We have added the error bars for CIFAR10, CIFAR100, and CIFAR100-LT. We will add the error bars for ImageNet in the final version.

---

> > ### Comment · Reviewer_yngU · 2021-11-28
> > **Response to Authors' Rebuttal**
> >
> > Thank you for the detailed response to my initial questions in the review.
> >
> > **Regarding triplet term**: I understand that $f_{triple}$ adds to the expressiveness of utility function, but it remains unclear if the pairwise function of (5) is insufficient to model diversity of samples in practice. I believe ablations studies and qualitative results (e.g. scatter plots suggested by reviewer 4ATq) are needed to justify the need for this objective.
> >
> > **Regarding balancing constraints**: I was not questioning the effectiveness of balancing constraint, but specifically the effect of introducing boundary-level balancing in addition to a naive class-level constraint. Again, I think numerical comparisons or visualizations would make the contribution much more convincing.
> >
> > **Regarding hyperparameters**: I appreciate the additional explanations, but would love to see further analysis on how values of important hyperparameters (e.g. $\lambda_i$ for each utility term) affect performance of learned models.

---

### Official Review · Reviewer_Cp2k · 2021-11-01

**Correctness:** 3
**Technical Novelty And Significance:** 3
**Empirical Novelty And Significance:** 2
**Recommendation:** 5
**Confidence:** 3

**Main Review:**

AFTER REBUTTAL:

I thank the authors for their detailed rebuttal and for testing out some of the suggested ablations. However, in light of the number of ablations and some of the confusion regarding the motivation behind class and boundary balancing, I feel this paper may not yet be ripe and could do with another round to review the explored experiments and revise the intuition and justification behind the setup. Therefore I leave my score as is.

-------------------


I like the review of how different strategies are biased to miss class and boundary diversity, as well as the matroid constraints for incorporating them. However, I feel that the paper can be improved with additional ablation and analysis. Below are some suggestions.


- I found the histograms in Fig 3 and corresponding discussion confusing. It would be useful to revise the axis and give some more information as to how a reader can interpret the plot. I interpret the x-axis of the histogram as “boundaries covered by $n$ images”. So for example with CIFAR10, there are 12 decision boundaries that are covered by only 1 image, and 13 boundaries covered by 2 images. Is this correct?
- It seems there is extensive parameter tuning involved, particularly in selecting how much to weigh the different terms in the objective function. Parameter tuning in active learning is a fundamental challenge since we typically don't have the validation data to tune parameters at onset. Can you provide details to how much parameter tuning was involved and how your custom objective would work in practice?
- An alternative to the above point, in terms of delineating the value of parameter tuning would be to implement the class and decision boundary constraints with respect to existing active learning strategies. For example, $\lambda_1 = 1, \lambda_2 = \lambda_3 = 0$ would give us a fair comparison on the value of class and boundary balance and how it improves a well-known existing method like margin.
- The improvement of the proposed method over baselines is attributed to the balancing constraints (e.g., comparing submod-bal vs submod). How well does the proposed method work in covering class and decision boundaries? For example, it may be useful to have a histogram of labeled class frequencies after each round of active learning for each method. Ideally, we expect the proposed method to select more balanced sets to label than baselines. We can do a similar analysis replicating Fig. 3 evaluating selections in each round with the decision boundary of a final classifier trained with 100% of the data as ground truth. These analyses can answer: how much does the current method improve over baselines in selecting class and boundary-balanced data, whether there is room for more improvement, and what is more important (class or boundary balance?)
- I am most interested to see what this looks like on CIFAR100-LT since this is where the proposed method shines the most and I expect class-balancing to be a key contributor. A good class balancing AL strategy can very much help imbalanced data collection practices and I encourage the authors to emphasize this value of their proposed method further. For example, another ablation could be to test on different imbalance factor for CIFAR100-LT and track the relative improvement of the proposed method over baselines as we vary the degree of imbalance.


**Summary Of The Paper:**

Most batch-mode active learning strategies involve maximizing a sub-modular score function of the value of each image to be labeled. This paper demonstrates that current methods fail to sample diverse classes or images near decision boundaries, arguably one of the most important regions to obtain label information. The paper proposes a technique for incorporating class-balance and boundary-balance constraints to the sub-modular optimization problem and performs experiments over several image data sets to show the value of class and boundary balancing.


**Summary Of The Review:**

Overall the approach and results look interesting, but the limited experiments make it difficult to fully capture the value of the methods proposed. I would be happy to increase my score if the authors can further expand numerically on where the improvements are coming from.

---

> ### Author Response · Authors · 2021-11-23
> **Response to Reviewer Cp2k**
>
> Thanks for the detailed feedback. We address the major concerns below.
>
> **Confusion regarding Figure 3.**
>
> We will clarify this better in the paper. In CIFAR10, we can have a maximum of 100 decision boundaries and not all of them will have images close to them. Around 90% of these decision boundaries have images associated with them. On the other hand we have only 15% of the decision boundaries with images in ImageNet. This is understandable since ImageNet can have up to million decision boundaries and not every pair of classes are adjacent to each other even in higher dimensions.
>
> In Figure 3, we just show the number of images associated with each of these decision boundaries. In Figure 3, we show that 12 out of the 100 decision boundaries have only one image each. Please note that each image is associated with only one decision boundary based on the top two class probabilities. So we are essentially talking about 12 different images associated with each of these 12 decision boundaries.
>
> **Parameter tuning**
>
> Reviewer 4ATq had the same question. Our uncertainty values are in the range [0, 1] and the cosine similarity values vary from -1 to 1. We used $k=10$. We set a larger value for the coefficient of margin compared to diversity since we have $k=10$. T_{thresh} is chosen to just provide some additional benefit from the triple clique without an aggressive search. We did not perform an elaborate hyperparameter search, but we just tested around 10 or fewer hyperparameter settings manually based on the relative importance of different criteria.
>
> **Ablation with respect to $(\lambda_1, \lambda_2, \lambda_3)$**
>
> We did an ablation study for various choices of lambdas. As we mentioned in the paper, the clique function shows an improvement of 0.49% in CIFAR and 1.13% in CIFAR100-LT dataset. This is obtained by setting lambda_3 = 0. We also tested the role of class and boundary balancing constraints by removing them from the optimization problem. The balancing constraints show an average improvement of 0.61% in CIFAR datasets, and 2.83% in CIFAR100-LT. As per your suggestion, we also studied the case where $(\lambda_1=1, \lambda_2=0, \lambda_3=0)$and observed that the proposed algorithm is around 0.62% better than Margin on CIFAR-10 and CIFAR-100. On the CIFAR100-LT, the proposed algorithm is around 4.1% better than Margin due to the class imbalance in these datasets.
>
> **Visualization before and after boundary constraint**
>
> We did perform this visualization and the constraints did achieve more balanced subsets. While this should happen naturally due to these balancing constraints, it is not that straightforward since we compute the association to the decision boundaries using the SEED model predictions and this is not based on true labels. In practice we did observe more balancedness even considering the true labels. Thanks again for this excellent suggestion.  We will be happy to add this and this will definitely improve the contribution and understanding of this paper.
>
> **More analysis on CIFAR100-LT**
>
> Thanks for the great suggestion.The balancing constraints show an average improvement of 0.61% in CIFAR datasets, and 2.83% in CIFAR100-LT. One can also use the balancing constraints individually for classes and boundary constraints. At a high level, class-level balancing constraint only looks at associating one class for every image, and boundary-level balancing constraint looks at associating two classes to every image. Please note that our formulation uses the intersection of matroid constraints for addressing class and boundary balancing constraints. We observed that class-level boundary constraints provided more improvement (2.13%) compared to boundary-level constraints (1.69%) on CIFAR100-LT.
>
> We can also generate more ablation by varying the degree of balancedness. This can be achieved by increasing the threshold for class and boundary association. Since we are working with predicted class labels and not the true labels, we did not pursue this direction. In general, we also wanted to study this with true labels although this violates the active learning principle. We should be able to generate all the additional visualizations and ablation study for all the datasets in the final version.

---

### Official Review · Reviewer_4ATq · 2021-11-03

**Correctness:** 3
**Technical Novelty And Significance:** 3
**Empirical Novelty And Significance:** 2
**Recommendation:** 3
**Confidence:** 4

**Main Review:**

### Strengths

S1) This paper identifies and addresses a common issue in active learning, which is that selection criteria generally don't address class imbalance and decision boundary imbalance.

S2) Experiments were performed on multiple datasets, including a highly class-imbalanced dataset (CIFAR-100LT) which may be more reflective of real-world settings.

### Weaknesses and Issues

**W1) Need to clarify misleading "constant approximation guarantee"**

The paper claims to have a "constant approximation guarantee," which is misleading. This guarantee is only with respect to the proposed selection criteria $f(S)$, but there is no guarantee that a set $S$ which maximizes $f(S)$ is actually a good training set. To the best of my knowledge, the correlation between $f(S)$ and test accuracy is merely heuristic and empirical. This point needs to be made much clearer in the text to prevent readers from conflating a "constant approximation guarantee" for maximizing $f(S)$ and the (non-existent) guarantee for maximizing test accuracy.

In contrast, active learning methods based on influence functions (see W2 below), actually try to directly maximize test set accuracy.

**W2) Missing discussions and comparisons against other related works**

The paper is missing discussions and comparisons against about other relevant active learning techniques such as [Deep Bayesian Active Learning (Gal et al., 2017)](https://proceedings.mlr.press/v70/gal17a) and [Active Learning via Influence Functions (Xu and Kazantsev, 2019)](http://arxiv.org/abs/1905.13183). Note that DBAL can be considered as another "uncertainty" metric. However, active learning via influence functions does not seem to fall within the uncertainty vs. diversity framework that this submission describes. Therefore, I would be particularly interested in seeing a discussion and comparison between the proposed framework and active learning via influence functions.

**W3) How does the proposed framework address the limitation that "no single subset selection criterion achieves the best performance"?**

The authors write, "To address the first limitation, we develop a unified algorithm based on maximization of a submodular function, which can combine different selection objectives." The limitation that they refer to is "First, there is no single subset selection criterion that achieves the best performance on different classification datasets."

It is unclear to me how the proposed framework addresses this limitation. Putting the various selection objectives into a single selection criteria function $f$ does not magically mean that the combined criteria achieves the best performance on different classification datasets. The authors' claim seems very misleading.

**W4) Need better justification and discussion about $f_{triple}$**

It is unclear to me how $f_{triple}$ is practically different from $f_{diversity}$. Are there situations where the two are very different? Is there a theoretical justification for why including both metrics is better than including just one of the two? Could you show a scatter plot of the two metrics (e.g., sample many different sets $S$, then plot $f_{diversity}(S)$ on the x-axis and $f_{triple}(S)$ on the y-axis)?

Also, I'm not sure I fully understand what the "volume consumed by the embeddings of the triplet $\{i,j,k\}$" means. A more thorough discussion of this is warranted. How is $\mathcal{T}_{thresh}$ chosen? And why use a threshold on $t(i,j,k)$, instead of a proportional penalty on $t(i,j,k)$?

**W5) How to select hyperparameters parameters like $\lambda_i$, $\gamma$, $\mathcal{T}_{thresh}$, and $k$**

I would appreciate a discussion on how to select the hyperparameters mentioned throughout the paper. The values chosen seem rather arbitrary.

**W6) What about active learning for regression?**

The paper only describes active learning for classification problems. The authors should make this clearer from the start of the paper. Are any of the ideas proposed here generalizable to regression settings?

**W7) Clarity Issues - please address each point individually in your response**

- In the "Subset selection and model evaluation" paragraph under "Section 3: Problem Statement", it is unclear what edges are included in $E$. For example, is $E$ directed? Is $E$ fully-connected? Only after reading through the end of the paper did I understand $E$ to be the edge set of a $k$-degree nearest neighbor, where $k$ is a hyperparameter. This point should be made clearer.

- Why define the margin score as $m(x) = 1 - \left( P(Y=bb|x) - P(Y=sb|x) \right)$? Why not define it more simply as $m(x) = P(Y=bb|x) - P(Y=sb|x)$?

- What unary function did you use for $f_{diversity}$ in your experiments? Did you use the constant term?

- Equation (7) says $M = (G, \mathcal{I}_p)$, but then defines $\mathcal{I}_c$. Looks like a typo.

- The authors write that on ImageNet, "we found that 15% of all possible boundaries were covered." Help me interpret this statement. Is 15% surprising? Is this more or less than the authors expected?

- In equation (10), what is $M$? It seems like $M$ is used both as an integer (for the number of decision boundaries) and as a matroid. Please clarify.

- In the "Evaluation" subsection under "Section 6: Experiments," the authors write, "Note that, on this dataset, accuracies with 80% and 90% subsets exceed that with the full dataset." Which dataset is this referring to?

- In the Ablation study subsection, what are the standard deviations for the reported improvement percentages? Are these relative improvements, or absolute improvements? On which dataset(s) did the authors test the larger values of $k$?

**Summary Of The Paper:**

The paper describes a new active learning algorithm for classification problems. The authors propose a selection criteria that trades off between uncertainty and diversity. The novelty lies in formulating a three-pronged approach to diversity that still manages to maintain submodularity of the selection criteria:
1) a traditional diversity metric based on cosine similarity of embeddings
2) a new triplet/clique "loss" that penalizes the selection of nearby points
3) class-balancing and boundary-balancing constraints

By showing this three-pronged approach maintains submodularity, they are able to devise a greedy algorithm that still has a constant approximation guarantee to the optimal selection subset (as measured by the selection criteria). Experiments on CIFAR-10, CIFAR-100, ImageNet, and CIFAR-100LT (long-tailed) suggest that the new approach performs as well as or better than state-of-the-art methods.

**Summary Of The Review:**

The active learning approach proposed by the paper builds incrementally on existing ideas of uncertainty and diversity sampling. They provide decent empirical evidence to suggest that their methods are at least as good as existing methods, and sometimes better. However, I hesitate to recommend this paper for acceptance as-is, due to misleading claims, incomplete discussions, and a number of clarity issues. Many of the design choices seemed arbitrary and require more thorough justification. I would be inclined to raise my score if the paper's current weaknesses and issues are properly addressed.

---

> ### Author Response · Authors · 2021-11-23
> **Response to Reviewer 4ATq**
>
> Thanks for the detailed feedback. Regarding your major concerns, we address them below.
>
> **W1) Need to clarify misleading "constant approximation guarantee"**
>
> Yes, the constant approximation guarantee is with respect to the objective function. We do not make any claim regarding the maximization of test set accuracy. Unfortunately, there is a big gap between theory and practice in active learning methods applicable to deep learning. Theoretical results are usually about how to achieve a sufficiently accurate model with a bound on the number of queries used.
>
> In this work, we are primarily interested in developing an objective function that performs well on large datasets such as ImageNet and has the flexibility to adapt to different criteria and constraints. While the main contribution of this paper is in the optimality guarantees and the formulation of the constraints, the submodular objective function is closely related to the k-center and classical uncertainty methods that have been theoretically studied in active learning settings.
>
> A recent work on k-center [Sener et al. 2017] shows that the minimization of the target risk corresponds to solving a k-center objective. However, note that this does not correspond to an objective function involving both diversity and margin. In fact, the authors of k-center [Sener et al. 2017] suggest this extension is an open problem. A recent work also looks at controlling the generalization error in submodular subset selection for linear regression and single-layer with ReLU settings in Sivasubramanian et al. 2021.  In the case of linear separators, margin sampling can use O(d log(1/epsilon)) labeled examples to get a model with error less the epsilon when the data lies in a ball of $R^d$ [ Balkan et al. 2007].
>
> **W2) Missing discussions and comparisons against other related works**
>
> Thanks for pointing to these references. We will be happy to cite both of these papers and add a discussion. GORAL is indeed a nice approach using an influence function. As you mentioned, the framework does not fall into uncertainty versus the diversity framework described in this paper. The utility or the unary function can utilize any function and we should be able to use the influence functions from GORAL as an additional criteria. It is nice to know that GORAL lends itself nicely to a batch-mode setting where the approximate batch utility is the same as the sum of the approximate individual utilities [Remark 2 in W2].
>
> **W3) How does the proposed framework address the limitation that "no single subset selection criterion achieves the best performance"?**
>
> Your observation is valid, but we want to clarify that there are methods out there that do not necessarily capture all the criteria such as diversity, uncertainty, and coverage. For example, k-center [Sener et al. 2018] only selects subsets that address coverage and diversity. In particular, they mention that incorporating uncertainty in the k-center objective would be beneficial and pose this as an open problem. The gold standard “margin” or uncertainty sampling methods do not use diversity, and diversity does indeed bring additional gain on some datasets. Through the use of the submodular objective, we can bring these different criteria into a unified objective function along with balancing constraints that are beneficial on long-tailed datasets.
>
> We agree with you completely. There is a little bit of parameter tuning, but we observed that the parameters we choose generalized to different datasets if the distance between the embeddings and the uncertainty values are in the same range.
>
> **W4) Need better justification and discussion about $f_{triple}$**
>
> The basic idea of the triple clique is straightforward and inspired from many vision papers. We can consider the original submodular objective function as a Boolean function with only pairwise terms as shown below:
>
> $f(x_1, …, x_n) = \sum_{i} a_i x_i + \sum_{i,j} b_{ij} x_i x_j + \sum_{ijk} c_{ijk} x_i x_j x_k$
>
> If the coefficients $a_i$, $b_{ij}$, and $c_{ijk}$ satisfy certain conditions the overall objective function is submodular. Mathematically, we can not find a function that uses only pairwise terms to approximate a function involving triple clique terms. Let us consider a simple example below:
>
> $f(x_1, x_2, x_3) = -x_1x_2x_3$
>
> We can have functions such as $g(x_1,x_2,x_3) = - \frac{1}{3} (x_1 + x_2 + x_3)$ that can match the triple clique function when $(x_1=0,x_2=0,x_3=0)$ and $(x_1=1,x_2=1,x_3=1)$, but not at for all combinations. While minimizing submodular functions, there are techniques that use additional variables to minimize the functions [Kolmogorov et al. PAMI 2004] by casting them as pairwise functions, but this is not generally not an easy thing to do. .
>
> Thanks. We used a simple function on triple cliques that is submodular. Great suggestion on using a penalty proportional to $t(i,j,k)$ instead of using a threshold on $t(i,j,k)$ for future work.

---

> > ### Author Response · Authors · 2021-11-23
> > **Response to Reviewer 4ATq (cont.)**
> >
> > **W5) How to select hyperparameters parameters**
> >
> > Our uncertainty values are in the range [0, 1] and the cosine similarity values vary from -1 to 1. We used $k=10$. We set a larger value for the coefficient of margin compared to diversity since we have $k=10$. T_{thresh} is chosen to just provide some additional benefit from the triple clique without an aggressive search. We did not perform an elaborate hyperparameter search, but we just tested around 10 or fewer hyperparameter settings manually based on the relative importance of different criteria.
> >
> > **W6) What about active learning for regression?**
> >
> > Yes, this paper does not consider active learning for regression. We will fix this in the draft. Nevertheless, we believe that the proposed method should generalize to regression setting, but not the balancing constraints. For your reference, a recent work looks at controlling the generalization error in submodular subset selection for linear regression and single-layer with ReLU settings [ Sivasubramanian et al. 2021]
> >
> > **W7) Clarity Issues**
> >
> > - E is obtained from the k-nearest neighbor graph for computational reasons. We consider an undirected graph and we use k=10 for all experiments. We will fix this in the paper.
> >
> > - We agree with your definition of margin score. Here we use utility values that are high for informative points and low for uninformative points and this also relates to the goal of maximizing the overall objective function.
> >
> > - Yes, we used the constant unary term.
> >
> > - Thanks for the careful reading .Yes, this should be I_c for denoting class balancing constraints.
> >
> > - 15% of the decision boundaries on ImageNet is significant and more than what we expected considering that we are talking about approximately 150,000 decision boundaries from 1000 classes. We will clarify this in the paper.
> >
> > - Thanks again for careful reading. We will use {\cal M} to avoid the confusion.
> >
> > - We have fixed the ambiguity regarding the 80% and 90% subsets. We observed that in all the datasets tested in this paper, we observed that 80% or 90% subsets, or both, were performing better or similar to the use of full datasets.
> >
> > - Ablation study subsection: We did all our ablation study on CIFAR(10 & 100) and CIFAR-LT datasets. The ablation on k was also conducted on CIFAR10 and CIFAR100 datasets.

---

### Decision · Program_Chairs · 2022-01-20

**Decision:**

Reject

**Comment:**

The paper studied an interesting yet challenging problem in active learning and provided an intuitive heuristic for selecting informative subset(s) of training examples. The reviewers generally find the paper well presented and highlight that the clarity of the exposition of the issues of existing query heuristics, especially for training deep models with class-imbalance data.

However, there are shared concerns among all reviewers in whether the existing experiments sufficiently justify the practical significance of the proposed heuristic (Reviewer 4ATq: Missing comparison against important baselines such as Gal et al 2017; Reviewer Cp2k: ablations of class and boundary balancing; Reviewer yngU: lack of comparison to SOTA and ablation for important hyperparameters; Reviewer oEcZ: lack comparison against SOTA). Reviewers also point out that the approximation guarantee is against an algorithm that is optimal wrt a somewhat ad-hoc objective, which makes the theoretical components of the paper not as significant. Given the above concerns, the paper does not appear to be ready for acceptance at the current stage.